# Exploring healthcare providers' experiences with specialty medication and limited distribution networks

Megan E. Peter[1], Autumn D. Zuckerman[1]*, Elizabeth Cherry[1‡], David G. Schlundt[2], Kemberlee Bonnet[2], Nisha Shah[1‡], Tara N. Kelley[1‡]

1 Department of Pharmaceutical Services, Vanderbilt University Medical Center, Nashville, Tennessee, United States of America, 2 Department of Psychology, Vanderbilt University, Nashville, Tennessee, United States of America

☯ These authors contributed equally to this work.
‡ EC, NS and TNK also contributed equally to this work.
* Autumn.Zuckerman@vumc.org

**Data Availability Statement:** We have provided the minimal data set requested. Each row represents a response from an interviewee and the

## Abstract

Integrated health-system specialty pharmacies (IHSSP) have shown high medication access, adherence, and provider satisfaction. The goal of this study was to explore healthcare providers' experiences with specialty medications distributed via Limited Distribution Networks (LDN) that do not include IHSSPs. We investigated healthcare providers' perceived impact of LDNs on clinic workflow, clinical practice, and patient outcomes. Interviews and focus groups were conducted with fourteen healthcare providers from four outpatient specialty clinics at an academic health system with an IHSSP. Qualitative analysis using an iterative inductive/deductive approach of coded transcripts was used to identify themes. Participants discussed requirements and barriers to communicating with insurance providers, drug manufacturers, and external pharmacies; time and effort required to navigate LDNs and impact on workload and clinic workflow; financial awareness of medication costs and methods for communication about financial information with patients; and advocating for patients to ensure access to necessary therapy and avoid missed doses or treatment lapse. Participants reported barriers to navigating LDNs that can interfere with clinic workflow and patient care. IHSSPs may reduce clinic burden by helping patients access, afford, and remain on therapy.

## Introduction

Specialty medications, typically high-cost medications that treat rare or complicated diseases, offer life-changing outcomes for patients. Manufacturer distribution models for specialty medications vary: some can be dispensed by traditional retail pharmacies and others can only be dispensed by accredited specialty pharmacies. Some drug manufacturers employ limited distribution networks (LDNs), mandating that the drug can only be dispensed by select specialty pharmacies included in the network. LDNs can help manufacturers ensure product integrity

coding used for analyses is found in columns A through M. The participant role and whether they were part of an interview or focus group are also shown.

**Funding:** The authors received no specific funding for this work.

**Competing interests:** The authors have declared that no competing interests exist.

**Abbreviations:** COREQ, criteria for reporting qualitative research; CTSA, Clinical Translational Science Award; IHSSP, integrated health-system specialty pharmacy; IQR, interquartile range; LDD, limited distribution drug; LDN, limited distribution drug network; MA, Master of the Arts; PhD, Doctorate of Philosophy; VSP, Vanderbilt Specialty Pharmacy.

and monitor its handling and administration, but also can benefit the manufacturer financially and may impede or delay medication access [1–3]. Patient access to specialty medications can be delayed by restrictions in insurance coverage or LDN requirements, which may create time-consuming and burdensome barriers for clinic staff to navigate [4, 5].

Many health-systems have developed integrated health-system specialty pharmacies (IHSSPs) to streamline patient access and improve adherence to specialty medications [6–8]. Some IHSSPs embed pharmacists and pharmacy technicians into ambulatory specialty clinics as part of the healthcare team to manage prescription access requirements. If the patient elects to use the IHSSP and the IHSSP is in network with both the drug manufacturer and patient's insurance provider, the IHSSP dispenses the medication to the patient. Integrated specialty pharmacists and pharmacy technicians monitor for therapy safety and effectiveness through clinic visits and virtual monitoring (telephone and telehealth communications). Patient outcomes and any barriers to medication access or adherence are communicated to prescribers through the electronic health record. However, IHSSPs cannot dispense medications or perform comprehensive medication monitoring unless included in manufacturer dispensing networks and insurance provider (payer) contracts.

Previous research has described high medication access and adherence in patients who fill medication from an IHSSP compared with limited distribution drugs filled from an external pharmacy [9–12]. Healthcare providers rate higher satisfaction with IHSSPs than external [13]. Moreover, health systems have voiced concerns regarding negative impact of LDNs, including fragmented patient care, higher clinic burden, and delayed patient access to therapy [1, 2, 14]. However, no known study has investigated healthcare providers' experiences with specialty medications in LDNs.

The goals of this study were to 1) explore healthcare providers' experiences with LDNs, particularly those specialty drugs an IHSSP does not have access to dispense due to manufacturer distribution restrictions (henceforth referred to as limited distribution drugs [LDDs]), and 2) to investigate healthcare providers' perceived impact of LDNs on clinic workflow, clinical practice, and patient outcomes.

## Methods

### Design and setting

This qualitative study was conducted at Vanderbilt University Medical Center, a large non-profit academic health system in the Southeast United States. As of July 2021, Vanderbilt Specialty Pharmacy (VSP) provides IHSSP services in 25 outpatient specialty clinical areas. Clinical pharmacists and pharmacy technicians help patients prescribed specialty medications access and afford therapy by assisting with insurance prior authorizations and appeals and linking patients with financial assistance programs when needed. If VSP is in network to dispense the specialty medication (based on manufacturer and payer contracts) and the patient chooses to fill medication from VSP, pharmacy staff will coordinate fulfillment and shipment to the patient. After patients initiate treatment, they are monitored through refill assessments, periodic pharmacist assessments, and clinic appointments. Per pharmacy accreditation guidelines, VSP uses robust clinical management protocols for specialty medications that can be dispensed. If patients use external specialty pharmacies for medication fulfillment, VSP pharmacists are unable to track medication access, fulfillment, and initiation. External specialty pharmacies maintain patient management protocols and follow legally mandated pharmacy dispensing requirements. Therefore, VSP cannot provide the same comprehensive services to patients managed by external specialty pharmacies to prevent duplication of efforts and patient confusion. This study was approved by the Vanderbilt University Institutional

Review Board. Methods were developed and are reported using the consolidated criteria for reporting qualitative research (COREQ) checklist (S1 Table. ISSM COREQ Checklist).

## Sampling

Purposive sampling was used wherein email invitations were sent to healthcare providers in five specialty clinics that prescribe a high volume of specialty medications. These clinics were selected to ensure participants were familiar with specialty medications that are dispensed through both the VSP and external specialty pharmacies. Participants were offered an interview or focus group, depending on convenience and preference.

## Data collection

After reviewing and signing a consent form, participants completed a brief survey assessing their demographic characteristics, current role in clinic, and years' experience with specialty medications. Semi-structured interviews or focus groups were then conducted, and follow-up questions were asked based on participants' responses (S1 Text. Semi-Structured Interview/ Focus Group Guide). Questions were developed by VSP pharmacists, leadership, and research team members/authors, then reviewed and annotated by Vanderbilt Qualitative Research Core personnel (not affiliated with VSP). Interview and focus group sessions followed a semi-structured guide focusing on the impact of LDNs on workflow, patient outcomes, and clinical practice. All interviews and focus groups were approximately 30 minutes in duration, took place at the workplace, and were audio recorded; the three-person focus group was video-recorded. A single female interviewer (MEP)who was a project manager within VSP, led all interviews and focus sessions. The interviewer holds a doctorate in behavioral sciences, and had minimal background in specialty pharmacy, no formal healthcare training, and no prior relationship with participants. The interviewer informed participants of her role at VSP and provided an overview of the study aims. All interviews were conducted at the workplace (Vanderbilt University Medical Center). Data saturation was not assessed as investigators determined a priori to perform all interviews/focus groups given the relatively small sample of clinics of interest based on use of LDD medications in the clinic.

## Analysis

Audio recordings of the interviews and focus groups were transcribed by student pharmacists and reviewed for accuracy and to remove all personal identifiers by the interviewer (MEP). The Vanderbilt Qualitative Research Core was contracted to complete coding and analysis. A hierarchical coding system was developed iteratively using the interview questions and two randomly selected transcripts (S2 Table. Coding System). There were six major categories: 1) Participant descriptive details; 2) Steps involved in prescribing; 3) Other people involved in the process; 4) Factors that influence medication prescriptions; 5) Facilitators and Barriers; 6) Specific medications mentioned, and 7) Suggestions. Each category was further subdivided into 2–13 subcategories with several subcategories refined to a third level in the hierarchy. Definitions were written for each category and a codebook was developed using Microsoft Excel (S2 Table). There were nofield notes to incorporate into the analysis.

The transcripts were imported into Excel spreadsheets and formatted using a coding template. Each line in the spreadsheet was a speaking turn. Separate columns indicated the role of the interviewee (e.g., nurse, physician), participant identification, speaker, and quote. Each quote could be coded using up to 13 codes. The two transcripts used to develop the coding system were coded independently by two coders who then compared their results. This process was used to further refine the coding system, add additional categories, and refine definitions.

The remaining nine transcripts were independently coded by two coders who reconciled their differences. The final coding consists of an agreement between two coders on how each quote should be characterized.

Qualitative analysis was performed using an iterative deductive/inductive approach [15, 16]. Deductively, authors used knowledge about clinicians and the health care system to structure our understanding of the coded quotes. Inductively, authors used the coded quotes to fill in details and to make connections between themes. Two qualitative researchers (PhD and MA not associated with VSP) reviewed the sorted quotes, identified themes, and flagged quotes for later use in presenting the data. Identification of themes was an iterative process starting with study questions and further modified by reviewing quotes. No interviews contradicted the overall conclusions about major themes.

## Results

### Participants

Fourteen providers from four specialty clinics participated in the study: a two-person focus group, a three-person focus group, and nine individual interviews. No providers refused to participate. As shown in Table 1, most participants were nurses (n = 6), White/Caucasian (n = 13) and women (n = 10). Median age was 37 years [interquartile range (IQR) = 33, 54], and participants had worked in their current role for a median of 4 years [IQR = 3, 7]. Transcripts were not returned to participants and participants did not provide feedback on the findings. No interviews were repeated.

Four themes were identified: communication, clinic workflow and workload, financial aspects of medication, and advocating for patients.

**Table 1. Characteristics of the sample (n = 14).**

| Characteristic | N (%) or Median [IQR] |
|---|---|
| Clinic | |
| Endocrinology | 3 (21%) |
| Hematology | 3 (21%) |
| Neurology | 4 (29%) |
| Pediatric Rheumatology | 4 (29%) |
| Job Title | |
| Physician | 4 (29%) |
| Nurse Practitioner | 4 (29%) |
| Nurse | 6 (43%) |
| Proportion of work time spent assisting patients with accessing specialty medicines | |
| Less than 25% | 8 (57%) |
| 25–50% | 6 (43%) |
| Time spent in current role, years | 4 [3.12–7.25] |
| Age, years | 37 [33–53.5] |
| Race | |
| Asian | 1 (7%) |
| White/Caucasian | 13 (93%) |
| Gender | |
| Female | 10 (71%) |
| Male | 4 (29%) |

## Communication

Participants discussed experiences communicating with health insurance providers, drug manufacturers, and external pharmacies, and compared communication regarding drugs dispensed by LDNs and non-LDNs.

**Communication with insurance providers.**   For most patients with prescription drug insurance, clinic staff communicate with the patient's insurance provider to approve coverage of medication. As demonstrated in the following exchanges, paperwork required by insurance providers can be cumbersome and, when not completed correctly, initiate back-and-forth between the clinic and insurance provider:

> At the point that I want to prescribe [a specialty medication] I would have to go to the provider room to look for treatment forms, find them, and then fill it out, which usually [is] pretty complicated. . ..like they have a patient assistance program or they have a nurse that [goes to the patient], and you check [a box on the form] sometimes for some conditions and not for others.

> (Nurse practitioner)

Another participant expressed challenges with communicating with insurance providers via phone.

> The hardest part of prior approval. . . is all the time on the phone [with insurance providers]. You could be on hold for 45 minutes, and that freezes you from doing anything else while you are waiting to speak to somebody to get whatever they request.

> (Nurse)

One participant expressed difficulty reaching an insurance representative with medical knowledge and challenges with receiving conflicting or inaccurate information from representatives at insurance providers.

> I know [with one specialty medication] there have been a lot of [patients with] lapses in therapy because of insurance where [the insurance provider] won't request all of the documents that they need in the beginning. And then because [the specialty medication] is [available] in two concentrations, a lot of times they will approve one concentration, but they haven't approved the other one, . . .The amount of time I have spent [communicating with insurance providers] the past two years is ridiculous. Just trying to get [someone on the phone], and the problem is you call these insurances and it's not even someone medical, so they have no idea what you are talking about when you are trying to explain, they just say, 'oh that's fine, I'll approve it.' I'm like, you didn't even approve the right [dose concentration]. Huge barrier there.

> (Nurse)

With an IHSSP, the clinical pharmacist and pharmacy technician are familiar with requirements for appropriate use of each medication; thus the pharmacy team can complete required paperwork efficiently and serve as a point of contact for insurance providers.

> A lot of [documentation and communication for LDDs] has now moved to the Vanderbilt Specialty Pharmacy with [the clinic specialty pharmacist]. She is doing a fantastic job with [specialty medication prescriptions]. She has taken over the certain meds that Vanderbilt

Specialty Pharmacy can source for the patient. . . that works very well for medications that Vanderbilt can source for the patient.

(Nurse)

**Communication with drug manufacturers.** Participants also interact with drug manufacturers, either to start patients on therapy or to enroll eligible patients in manufacturer free drug assistance programs which may provide free medication to uninsured or underinsured patients.

One nurse described the complex process of communicating with a manufacturer to start a patient on treatment and the challenges that the back-and-forth correspondence placed on clinic workload.

Sometimes forms would be faxed with a script written on them but the doctor didn't sign it. . .That would trigger a fax to come back, and it would. . . get in a stack with other faxes. So, you've got a stack of faxes. . . If one little thing is not on the prescription or not on the form like one box is not checked, it generates phone calls back. . .. Multi-page forms, different signatures, different things needed on there. . . The point is this is a complex role and it really needs a single point contact within the clinic. When it is just part of the routine clinic workflow and everybody is handling it, there are too many hands in the pie, things get dropped, boxes don't get checked, and it creates an incredible amount of work and it really delays patient care.

(Nurse)

Manufacturers eligibility criteria and coverage rules for free drug programs that provide medication for patients initiating therapy or experiencing insurance lapses may vary. Knowing these criteria and the process of patient enrollment can be challenging. When IHSSPs are in the distribution network, the team can sometimes facilitate free medication storage, access, and fulfillment using a clinical management protocol. The IHSSP does not maintain protocols for LDN medications, so clinic staff and the manufacturer free drug assistance programs must coordinate patient enrollment and medication fulfillment and shipping for uninsured or underinsured patients prescribed LDDs. As discussed by the following participant, clear information about patient eligibility for these programs is not always available, which can cause patients to miss doses and experience severe complications:

In August, [the manufacturer free drug assistance program] [told] us [a patient receiving an LDD] is not eligible for [the manufacturer free drug assistance program] anymore because he has exceeded the maximum amount, which that was never communicated to us that there was a maximum amount because those changes [to assistance program eligibility] occurred. . . It was a lot of phone calls between me, [the manufacture free drug assistance program], and in the meantime, there was another pharmacy [added to the process]. Things were not straightforward. He did end up missing a dose a week for 3 weeks and required hospitalization.

(Nurse)

## Communication with external pharmacies

When the IHSSP is not in network, prescriptions for the LDD must be sent to an external pharmacy to dispense medication to the patient. Participants expressed barriers to

communicating with external pharmacies, who may not inform the clinic staff about delays or barriers to medication coverage.

The following exchange demonstrates communication delays with external pharmacies, which in turn prolongs insurance approval:

> When [the clinic] send[s] a prescription electronically to [an external] pharmacy, usually unless the [external] pharmacist is going to pick up the phone and call us, we won't know about any problems for at least a week to maybe even two weeks, because [the external pharmacy] will send a fax. Then the fax goes into the fax queue and it gets pulled off [by clinic staff] but then gets distributed by provider. It gets put into our [Clinic] baskets, if we are in satellite clinics, we don't get it for a couple of days. So, there can be a week-long delay just to find out medicine needs prior authorization or needs a refill or any of those simple procedures.
>
> (Nurse practitioner)

In other cases, an external pharmacy may not have a medication in stock but does not communicate this with the clinic staff, leading to delays or periods of treatment lapses.

> [Before we knew a drug was limited distribution] we were sending the prescription to the patient's pharmacy, and we weren't getting calls back for quite some time that [clinic prescribers] couldn't prescribe the drug because the pharmacies were trying to get the drug. It did lead to a delay in treatment when we [Clinic] weren't using our [VSP] pharmacists to [assist with insurance prior authorization] because the communication was just so poor from the other pharmacies.
>
> (Nurse practitioner)

> [There is a] patient I have who is on [an LDD], there has been times when that limited distribution pharmacy didn't have the medicine either. [The external pharmacy] had to order it and so it delayed it by another week. . . Because that specialty pharmacy didn't communicate with us that they were having a delay in shipment, and we were told that the LDD pharmacies had the medicine on hand, which isn't always accurate either. [The external pharmacy] didn't communicate back to the family or to us. It basically came down to where [the patient was] a week late for [an] injection that can cause this patient to have [a] flare and joint disease.
>
> (Nurse practitioner)

When the IHSSP has access to dispense a drug, these communication barriers are alleviated because the integrated pharmacist communicates with patients and clinic staff about medication barriers.

> By [the clinic] being able to use VSP [to dispense the medication], it has cut down on that communication gap. Because, first, [VSP clinical pharmacists] are on the floor with us [in the clinic] so if there is a significant problem, the [VSP clinical pharmacist] will just come speak to us directly. Second, the [VSP clinical pharmacist] are able to use the message basket system and let us know immediately whether a patient's insurance has lapsed, if [the patient] can't get coverage, if it's too soon to refill, if [the patient is] not refilling it appropriately, if the family has concerns or questions regarding the medicine, we [Clinic] are able to know that through VSP immediately. Which helps [the clinic] then communicate with the families quicker, get the situation resolved, and get the patient back on track.
>
> (Nurse practitioner)

## Clinic workflow and workload

Communication and administrative barriers associated with LDDs can interfere with clinic workflow and other patient care responsibilities.

Nurses discussed how communication barriers associated with LDNs interfered with their ability to complete other clinical tasks:

> [Contacting manufacturer hub] would become very time consuming because then it requires calling the family, the hub, the insurance provider, calling the pharmacy because none of those people were communicating and we would become the point of contact and that was taking away from the flow of care in clinic.
>
> (Nurse)

A physician participant also shared how their workflow and schedule were impacted when nurses were required to manage LDD prescriptions:

> [Paperwork] was affecting us because that nurse was not available. . . all the time, so I would have to ask the patient to be in a waiting room. I would have to go to the workroom, call the nurse; the nurse would then come down. They had the paperwork that required my signature as well, so it did affect my schedule because I didn't know when the nurse was going to come. Sometimes the patients can wait and sometimes they could not wait, and then all of this was happening over mail and faxes. Again, circling back to me for signatures. So definitely much more inconvenient back then [before VSP integration].
>
> (Physician)

With IHSSPs managing these prescriptions, participants reported being able to spend more time counseling, educating, and supporting patients and their families.

> [The IHSSP] has given us so much more time back with the patients and that impacts the care they get. We have more time to talk about lots of different aspects of their care instead of spending extended amounts of time on hold with insurance providers, not even with talking with people who are able to make decisions for the patient. So, we are extremely appreciative of that. . . I think we have more time to triage parents and are getting calls back faster than they were able to before.
>
> (Nurse)

## Financial knowledge and communication

Participants shared their knowledge and experiences with high-cost medications and how they discuss financial information with patients.

**Financial knowledge.** Participants consistently mentioned the high costs of specialty medications. With LDDs, clinicians and clinic staff may not be aware of all available financial assistance resources nor have time to help patients navigate these programs.

> Most of the drug companies have some sort of patient assistance programs, but they are very different from one medicine to the next. Even within one medicine, they change their programs. So, it is very hard for me I never know who has this co-pay card or who has that [copay card].
>
> (Physician)

When the IHSSP can dispense a drug, a dedicated pharmacy team helps patients who express need access financial assistance programs, which participants found helpful for themselves and the patients:

Specialty pharmacists have been very helpful at accessing grants for [patients in the Medicare coverage gap] or talking to the drug companies themselves to get compassionate use for the drugs so these patients can continue on it. It used to be where our social worker or myself or the nurses had to do all of that and we just didn't have time. So, I think having the specialty pharmacy do it is extremely helpful, and it helps patients too.

(Nurse practitioner)

I know that pharmacy is going. . . to be able to help them with copay cards. It's a big stress on family when we tell them that we are starting this medicine that is going to be expensive, but then we can give them recourses to connect them with different available co-pay programs. Pharmacy is great for that because I never know. I know [manufacturers probably have a co-pay card] but you have to call and give them the website to check or call their patient assistance program and check.

(Physician)

**Financial communication.** Participants also shared their practices for communicating with patients about medication costs and the benefits of the integrated pharmacist providing transparency regarding treatment costs for patients. With the integrated model, prescribers felt they could tell patients upfront about typical insurance/financial aspects of specialty medications, then refer them to the IHSSP team who can help navigate financial assistance if needed. Prescribers felt confident when the IHSSP can dispense the drug because they trusted that patients will receive support needed to overcome financial barriers and initiate therapy.

One physician felt that patients are more comfortable discussing financial information with the integrated pharmacist than with their healthcare provider:

I can tell you in general patients will never voice their financial concerns to the physician. They are more open in doing that with a pharmacist or nurse or a social worker. So, usually what I tell patients is that these medications are expensive, when we send the message to the pharmacist, they will walk you through everything and give you kind of what your options are. . .. So, I sort of give them a little bit of a prelude to that so they are not caught unaware. But VSP team really owns all of that for us.

(Physician)

This physician also introduced patients to the clinical pharmacist, who works alongside clinic staff, so patients have a single point-of-contact:

I let them know they will be contacted by [the clinic pharmacist]. I try to have [patients] actually meet [the integrated clinical pharmacist], either before the process and sometimes there's logistical things to have them sign, or once they are on the therapy it's nice to have a face with the name. . .. [The pharmacist] is helping these people and they usually want to thank her for all of the work.

(Physician)

The following participants discussed how, when the IHSSP can dispense the drug, they can refer patients to the clinical pharmacist who oversees the logistical steps of accessing medication.

> [After deciding to prescribe a medication] I'll message [the clinic pharmacist]. . .she will look into patient assistance programs because she knows about them, so in those situations, I can say to a patient, "I'd like to try [a specific drug], but it's very expensive, and so that is prohibitive for some patients from using it, but it totally depends on your insurance and your financial situation, and there's a lot of different options so I'm going to send it to our specialty pharmacist, who knows about all that stuff, and she will let me know what the options are, and then you guys can decide if you're able to afford it."

> (Nurse practitioner)

## Advocacy

Regardless of the drug's network distribution status, participants prioritized obtaining the most appropriate drug for the patient, as determined by medical literature and/or patient needs or preferences, and, to the extent possible, expressed willingness to devote time and resources to ensure patients' needs are met.

**Choosing the right medication.**   Sometimes, clinicians have only one or few drugs to treat a patient's condition, with no or few alternatives available and must do what is necessary to help patients access treatment.

> They need the medicine they need and we just have to get it. I mean. . . whatever hoops we have to jump through to get that. It's not like I have [multiple options] to choose from. . . in [some drug classes] you have a little bit more of a choice but not really, there are still differences that make individual ones be preferred in cases and I mean I am going to pick which ever one is best medically, we just would have to deal with the logistics of it. . .

> (Physician)

One participant shared her willingness to 'jump through hoops' to ensure patients can access and afford medication:

> We [clinic] pick the best drug for the patient in the situation they are in. If that means [the clinic has] to jump through 30 hoops to get that medication for [the patient], we will. Because, you know, in our practice it is the patient, they are what we are here for. They're the reason why I am such an advocate for this, that drugs shouldn't be limited in distribution mainly for the price issue. These families are already sometimes struggling to decide whether I come to the doctor versus buy groceries.

> (Nurse practitioner)

Participants expressed empathy and concern for whether patients could afford medication but mentioned that for drugs that can be dispensed by the IHSSP, they trust that the clinic pharmacist will direct patients to financial assistance programs. This reduced provider concerns about treatment affordability and access.

> I don't want to sound naïve, but I still want to go for the best [medication for the patient]. Having said that, of course if a patient cannot afford it [I take that into account], but I still

try for the first option clearly what I think is the best regardless of the cost and the logistics because we have such great support for the logistics.

(Physician)

**Coordinating care to prevent medication access gaps.** Nurses devoted time and effort to ensure that patients maintain access and adherence to therapy. When LDDs must be initiated during a hospitalization nurses ensure that paperwork has been completed so the patient has access to the medication after discharge:

When [the patient is prescribed a specialty medication while] inpatient, [the inpatient pharmacy] is able to give the medication inpatient. . . We just need to make sure before [the patient goes] home that a prior authorization has at least been started, and then [the manufacturer free drug assistance program] can send out the [free drug], but we also need to know when that is going to be sent out.

(Nurse)

The IHSSP maintains clinical patient management protocols for medications that are stocked and dispensed by the IHSSP, including sample medications, which are dispensed when deemed appropriate by prescribers to trial safety and/or efficacy. IHSSP staff gain expertise in navigating available patient assistance programs for medications dispensed.

[Whether patient's health declines while waiting for drug approval] depends on the medication. Like with [non-LDD], we supply [non-LDD], so we can give you a sample until we wait the two months when insurance [will approve/cover the medication] With the [LDD] it's not [dispensed by] our pharmacy, there's a whole other playing field there.

(Nurse)

## Discussion

This study details healthcare providers' experiences with limited distribution specialty medications at an academic health center with an IHSSP. Healthcare providers in five specialty clinics discussed the challenges associated with LDNs, and the benefits of an IHSSP, specifically that IHSSPs enable providers and clinic staff to better focus on clinical care, streamline the patient's access to medication, and provide essential financial assistance support that allow for confident specialty medication prescribing, and better patient outcomes. This is the first known study to qualitatively explore healthcare providers' experiences with LDNs and investigate how LDNs influence providers, patients, and clinics. This study builds on previous reports that demonstrate the impact of LDDs on time to medication access, health system expenses, administrative and financial burden for clinics, and fragmentation in care [1–3, 10–12]. Understanding specialty providers' perspectives on the perceived impact of LDNs is necessary for specialty pharmacy stakeholders to help improve efficient, equitable treatment for all patients.

Participants discussed the delayed and fragmented communication with external parties when prescribing a LDD, which interferes with patient's ability to access medication. Previous studies have reported that for each specialty medication prescription, clinic staff spend approximately two hours completing paperwork requirements and coordinating with insurance providers and external specialty pharmacies [17]. One study reported that half of oncology prescriptions required five or more phone calls by clinic staff before the patient obtained the

prescription [5]. If a drug can only be distributed by select pharmacies—as mandated by either the manufacturer LDN or the pharmacy benefit manager's pharmacy restrictions—prescribers must send the prescription to a specific external pharmacy; this may be unknown to prescribers upfront [4, 18]. These requirements can delay patient access to medication, or cause patients to abandon the prescription, sometimes resulting in negative clinical outcomes for patients with specialty diseases [5, 18]. In our study, participants expressed that communication barriers were reduced when a drug could be dispensed by the IHSSP. This qualitative data corroborates previous findings showing that access time is significantly longer for specialty medications with LDNs compared with non-limited distribution, and that medication access time is reduced after the IHSSP is included in the LDN for multiple sclerosis, hematology, and neurology clinics [10–12]. Conversely, when a LDN that excluded the IHSSP was imposed, one health system found delays in drug acquisition and more variable and unreliable medication delivery times [2].

Participants also discussed barriers to patient adherence to LDDs, such as when the external pharmacy is out-of-stock of the medication, or the patient's insurance benefits change. After patients initiate prescribed therapy, delayed and fragmented communication can cause patients to miss doses, which may cause urgent healthcare needs. When drugs can be dispensed by the IHSSP, the pharmacy team conducts monthly assessments, and prescribers are notified if patients report an issue with medication (adverse event, worsening symptoms) or encounter barriers to medication access, facilitating close monitoring and real-time communication with providers to address concerns or barriers. If the IHSSP is included in the LDN, clinical patient management protocols are in place and pharmacy staff have expertise in determining patient eligibility and enrollment requirements for free drug programs. Samples can also be stocked and dispensed as deemed appropriate by the prescriber for the patient to try the medication and evaluate safety and effectiveness. Previous studies have reported high medication adherence in patients who fill medication from an integrated pharmacy [7, 19–21] and discussed how clinical pharmacists at integrated specialty pharmacies can help patients navigate transitions in care to ensure medication adherence [22].

Many participants discussed the high cost of specialty medications and the impact of financial barriers on patient outcomes. Patients subject to high medication costs have lower adherence and experience financial toxicity—referring to distress and financial problems resulting from high medication costs—which is associated with worse patient outcomes and quality of life [23–25]. Participants felt confident prescribing medications when the IHSSP can dispense, as they trusted the pharmacy team would help the patient access treatment. Healthcare providers also discussed the benefit of a specialty pharmacist in clinic who could communicate with patients about financial assistance options. Patients filling medication from an IHSSP often pay low out-of-pocket medication costs due to the extensive knowledge surrounding available assistance programs for patients who express financial need [9, 19–21]. This in turn may lead to lower financial toxicity in patients who require specialty medication [23]. When patients lose or change insurance coverage, integrated specialty pharmacists coordinate care to ensure patients can obtain affordable medication during this transition [22]. IHSSPs often have access to internal safety net programs that can assist patients who qualify when other options are not available.

Despite the challenges and barriers associated with LDNs, participants prioritized advocating for their patients, expressing willingness to conduct extra work to identify and enroll patients in financial assistance programs when the external specialty pharmacy fails to provide adequate support to the extent possible to ensure patients access the best medication. These tasks, however, can interfere with other clinic responsibilities and patient care. When medications can be dispensed by the IHSSP, the pharmacy team oversees tasks needed for the

prescriptions; thus, nursing staff have more time to spend with patients. IHSSP teams often must assist patients prescribed an LDD who are having issues obtaining medications from external pharmacies. However, lack of transparency into external patient pharmacy records and lack of knowledge of the operational failures causing the gap in patient care can make this process challenging. Additionally, IHSSP efforts to address external pharmacy issues substantially increase workload and are unable to be quantified to support adequate staffing models.

Given our study's findings and other emerging research, it is worth exploring the rationale of LDNs without established and transparent criteria for network access. Drug manufacturers cite several reasons for using LDNs, including the ability to control medication product integrity, to closely monitor the medication's handling and administration, to ensure patients receive disease management programs from specific specialty pharmacies, and to more easily ensure FDA requirements are met [1–3]. Manufacturers also realize financial benefits of LDNs, through closer inventory management, lower distribution fees, and higher revenue generation [1]. Manufacturers participating in Medicaid and Medicare must provide outpatient drugs to eligible health care organizations and covered entities at reduced prices under the 340B program. Allowing IHSSPs with capabilities that meet and can often exceed the criteria for inclusion into an LDN fulfills this requirement. Several healthcare providers in our study voiced the benefits of IHSSPs when managing specialty medications and quantitative data from previous literature demonstrates IHSSPs can provide the high-touch care delivery needed to ensure safety, administration, and monitoring requirements are met. Therefore, allowing eligible IHSSPs that have robust capabilities fulfills manufacturer distribution goals while ensuring a streamlined patient journey and optimal outcomes.

## Future directions

Given the specific requirements for storing, handling, distributing, and monitoring the safety of specialty medications, the intent behind LDNs is reasonable. However, as noted by participants in this study, IHSSPs may augment manufacturers' goals, including the ability to mirror high touch clinical trial conditions by helping patients afford, initiate and maintain on specialty medication. Recently, the American Society of Health System Pharmacists released recommendations from the Specialty Pharmacy State of Practice in Hospitals and Health Systems Future Directions Summit which included, "hospital and health-system specialty pharmacies should develop strategies with payers and manufacturers that promote access to and establish criteria for limited distribution networks, such as value-based contracting, cost-management strategies, medication affordability assistance, and care coordination" [26]. A structured collaboration between manufacturers and IHSSPs could allow the manufacturer to minimize costly resources that lead to optimal patient outcomes (e.g. education, ongoing monitoring, financial assistance) as these are often core services of an IHSSP. Additionally, better IHSSP/manufacturer collaboration and data sharing could enable the manufacturer to gain insight into the streamlined patient journey facilitated by IHSSPs, and provider perspectives on the benefits of this model. Introducing transparent, tailored criteria for inclusion into LDNs, along with data sharing agreements, could help ensure better use of the restricted network. More research is needed to understand specialty pharmacist and manufacturer perspectives on the impact of LDNs and potential solutions for improving access while maintaining patient safety.

## Limitations

Participants were healthcare providers, mostly nurses with similar demographics, practicing within outpatient specialty clinics at single academic healthcare center with an IHSSP. Thus,

experiences with specialty medications and LDNs may not generalize to all healthcare providers and care settings. The sample size of the study was small, but participants represented a range of specialty clinics and healthcare professions, and participants had experience working with internal and external specialty pharmacies. Larger focus groups would have allowed for richer data, however including more participants was limited by few clinicians/staff having the knowledge and experience needed to speak to the topic. Participants were individually selected by investigators based on their experience with prescribing or coordinating care for specialty medications that are distributed through an LDN; this recruitment method may have introduced some unknown selection bias. The investigators did not plan for interviewees/focus group participants to review or confirm the interview transcripts, which could impact data quality.

Having a non-health professional interviewer may have been beneficial to reduce potential bias as she did not have personal experience with LDNs and was able to approach the interview/focus group without assumptions regarding the LDN process. However, the lack of healthcare experience may have limited her ability to ask more comprehensive in-depth questions due to limited experience with the topic.

## Conclusion

This study reports perspectives from specialty providers and clinic staff on the impact of limited distribution drug networks in real-world practice. As noted by participants, IHSSPs reduce clinic burden by helping patients access, afford, and remain on therapy. When IHSSPs are included in a drug's LDN, healthcare providers report high satisfaction, more confidence in prescribing specialty medications, improved workflow efficiency, and better patient outcomes. Manufacturers who wish to distribute specialty medication through LDNs should create and communicate transparent criteria for specialty pharmacies to meet distribution goals.

## Supporting information

**S1 Table. ISSM COREQ checklist.**
(PDF)

**S2 Table. Coding system.**
(DOCX)

**S3 Table. Coded minimal data set.**
(XLSX)

**S1 Text. Semi-structured interview/focus group guide.**
(DOCX)

## Acknowledgments

We would like to thank Heather Laferriere and the Annette and Irwin Eskind Family Biomedical Library staff for literature review assistance. We acknowledge Jacob Jolly, PharmD for his contribution to study design and manuscript review.

This project described was supported by CTSA award No. UL1 TR002243 from the National Center for Advancing Translational Sciences. Its contents are solely the responsibility of the authors and do not necessarily represent official views of the National Center for Advancing Translational Sciences or the National Institutes of Health.

## Author Contributions

**Conceptualization:** Megan E. Peter, Autumn D. Zuckerman, Elizabeth Cherry, Nisha Shah, Tara N. Kelley.

**Data curation:** Megan E. Peter, Autumn D. Zuckerman, David G. Schlundt, Kemberlee Bonnet.

**Formal analysis:** Megan E. Peter, David G. Schlundt, Kemberlee Bonnet.

**Investigation:** Megan E. Peter, Autumn D. Zuckerman.

**Methodology:** Megan E. Peter, Autumn D. Zuckerman, Kemberlee Bonnet, Nisha Shah.

**Project administration:** Megan E. Peter, Autumn D. Zuckerman.

**Resources:** Megan E. Peter, Autumn D. Zuckerman, Tara N. Kelley.

**Supervision:** Megan E. Peter, Autumn D. Zuckerman, Tara N. Kelley.

**Writing – original draft:** Megan E. Peter, Autumn D. Zuckerman, Elizabeth Cherry, David G. Schlundt, Kemberlee Bonnet.

**Writing – review & editing:** Autumn D. Zuckerman, David G. Schlundt, Kemberlee Bonnet, Nisha Shah, Tara N. Kelley.

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
