## [Decision Letter · Decision Letter 0]

18 Feb 2022

PONE-D-21-28656Exploring healthcare providers' experiences with specialty medication and Limited Distribution NetworksPLOS ONE

Dear Dr. Zuckerman,

Thank you for submitting your manuscript to PLOS ONE. After careful consideration, we feel that it has merit but does not fully meet PLOS ONE’s publication criteria as it currently stands. Therefore, we invite you to submit a revised version of the manuscript that addresses the points raised during the review process.

We look forward to receiving your revised manuscript.

Kind regards,

Wilfred Njabulo Nunu

Academic Editor

PLOS ONE

https://journals.plos.org/plosone/s/fileid=ba62/PLOSOne_formatting_sample_title_authors_affiliations.pdf".

2. In the Methods section of the manuscript, please consider including more information on the number of interviewers, their training and characteristics; and please provide the interview guide used.

5. Please upload a copy of Supporting Information Table 2 which you refer to in your text on page 5.

Reviewers' comments:

Reviewer's Responses to Questions

**Comments to the Author**

1. Is the manuscript technically sound, and do the data support the conclusions?

Reviewer #1: Yes

2. Has the statistical analysis been performed appropriately and rigorously? 

Reviewer #1: Yes

3. Have the authors made all data underlying the findings in their manuscript fully available?

Reviewer #1: Yes

4. Is the manuscript presented in an intelligible fashion and written in standard English?

Reviewer #1: Yes

5. Review Comments to the Author

Reviewer #1: We have reviewed the study reporting perspectives from specialty providers and clinic staff on the impact of limited distribution drug networks in real-world practice. The authors have shown that participants noted that IHSSPs reduce clinic burden by helping patients access, afford, and remain on therapy. When IHSSPs are included in a drug’s LDN, healthcare providers report high satisfaction, more confidence in prescribing specialty medications, improved workflow efficiency, and better patient outcomes. Manufacturers who wish to distribute specialty medication through LDNs should create and communicate transparent criteria for specialty pharmacies to meet distribution goals. We find the study contributing to the knowledge in service delivery in healthcare. We have included the MS highlighting areas we think may need to be corrected or included. Key words and abbreviations may be included in the MS as well.

6. PLOS authors have the option to publish the peer review history of their article (what does this mean?). If published, this will include your full peer review and any attached files.

Reviewer #1: **Yes: **Prof Dr Mavondo, Greanious Alfred

---

## [Author Response · Author response to Decision Letter 0]

15 Mar 2022

RESPONSE: 

We appreciate this reminder and have reviewed our manuscript to ensure it meets styling requirements. The template was appreciated. 

File naming updated:

S1 Table 1. ISSM COREQ Checklist

S2 Text. Semi-Structured Interview.Focus Group Guide

S3 Table 2. Coding System (updated number to reflect new supplementary material provided)

File added with same naming format:

S4 Table 3. Coded Minimal Data Set

2. In the Methods section of the manuscript, please consider including more information on the number of interviewers, their training and characteristics; and please provide the interview guide used.

RESPONSE:

We have modified the interviewer description to be more detailed and clear (page 5, lines 108-112).

We have uploaded the semi-structured interview and focus group guide (S2 Text. Semi-Structured Interview.Focus Group Guide).

RESPONSE:

We have uploaded a minimal data set that was used for coding and analysis. Please let us know if this is not acceptable.

4. Please upload a copy of Supporting Information Table 2 which you refer to in your text on page 5.

RESPONSE:

I confirmed that this information was uploaded in the original submission. It is now labeled S1 Table 1. ISSM COREQ Checklist. Based on the journal guidelines, we thought this name was acceptable, but we did remove the Complete and the underscores please let us know if there is a preference for a different name. Guideline excerpt:

Authors may use almost any description as the item name for a supporting information file as long as it contains an “S” and number. For example, “S1 Appendix” and “S2 Appendix,” “S1 Table” and “S2 Table,” and so forth. 

5. Please review your reference list to ensure that it is complete and correct. If you have cited papers that have been retracted, please include the rationale for doing so in the manuscript text, or remove these references and replace them with relevant current references. 

RESPONSE: We have reviewed the reference list and confirm they are complete and correct. We updated reference #26 as this data has now been published

6. We have reviewed the study reporting perspectives from specialty providers and clinic staff on the impact of limited distribution drug networks in real-world practice. The authors have shown that participants noted that IHSSPs reduce clinic burden by helping patients access, afford, and remain on therapy. When IHSSPs are included in a drug’s LDN, healthcare providers report high satisfaction, more confidence in prescribing specialty medications, improved workflow efficiency, and better patient outcomes. Manufacturers who wish to distribute specialty medication through LDNs should create and communicate transparent criteria for specialty pharmacies to meet distribution goals. We find the study contributing to the knowledge in service delivery in healthcare. We have included the MS highlighting areas we think may need to be corrected or included. Key words and abbreviations may be included in the MS as well.

RESPONSE:We appreciate the reviewer’s comments and feedback on the manuscript. Suggested grammatical edits in the attached document that was provided have been incorporated in the updated manuscript. We have carefully reviewed the manuscript and added additional grammatical changes where appropriate.

---

## [Decision Letter · Decision Letter 1]

14 Jun 2022

PONE-D-21-28656R1Exploring healthcare providers' experiences with specialty medication and Limited Distribution NetworksPLOS ONE

Dear Dr. Zuckerman,

Thank you for submitting your manuscript to PLOS ONE. After careful consideration, we feel that it has merit but does not fully meet PLOS ONE’s publication criteria as it currently stands. Therefore, we invite you to submit a revised version of the manuscript that addresses the points raised during the review process.

We look forward to receiving your revised manuscript.

Kind regards,

Jonas Bianchi, DDD, MS, Ph.D

Academic Editor

PLOS ONE

Reviewers' comments:

Reviewer's Responses to Questions

**Comments to the Author**

1. If the authors have adequately addressed your comments raised in a previous round of review and you feel that this manuscript is now acceptable for publication, you may indicate that here to bypass the “Comments to the Author” section, enter your conflict of interest statement in the “Confidential to Editor” section, and submit your "Accept" recommendation.

Reviewer #1: (No Response)

Reviewer #2: (No Response)

2. Is the manuscript technically sound, and do the data support the conclusions?

Reviewer #1: Yes

Reviewer #2: Yes

3. Has the statistical analysis been performed appropriately and rigorously? 

Reviewer #1: Yes

Reviewer #2: N/A

4. Have the authors made all data underlying the findings in their manuscript fully available?

Reviewer #1: Yes

Reviewer #2: Yes

5. Is the manuscript presented in an intelligible fashion and written in standard English?

Reviewer #1: Yes

Reviewer #2: Yes

6. Review Comments to the Author

Reviewer #1: We have made comments to the Authors in our initial review where we had accepted the MS for publication.

Reviewer #2: Thank you for the opportunity to review this manuscript. Overall, the manuscript reads well and is easy to follow. However, the following points need to be addressed before it gets published.

Methods:

It is unclear how the authors ensured data saturation, which is an important process for deciding how many people to interview in qualitative studies

It appears that the interviewee selections are mainly based on convenience rather than purposive sampling. Ten out of 14 participants (71.5%) are nurses/nurse practitioners. As such, the findings mostly represent nurses’ perspectives rather than the broader healthcare providers perspectives.

Although clinical pharmacists are the main players/stakeholders in LDN/VSP, the authors failed to incorporate pharmacists’ perspectives in the study, which is a missed opportunity to better understand the complexity of the IHSSP and LDN systems. Also, it would have been better to incorporate manufacturers’ perspectives. Unless we hear all stakeholders’ views, it would be difficult to come up with workable solutions.

It is unclear why the authors didn’t incorporate the field notes in their data analysis.

“Member check” is an important step in ensuring the trustworthiness/quality of qualitative data. However, the authors didn’t provide an opportunity for the interview to comment on the findings or edit/confirm the interview transcripts.

The focus group discussions only included two or three people. It is usually not recommended to have FGDs with less than five or six participants.

The authors should also discuss the pros and cons of a non-health professional interviewer. Although having a non-health professional as an interviewer may help to avoid bias, there are several limitations of this approach in health research, and this should be properly discussed in the discussion section.

Line 576-578: “Participants were individually selected by investigators based on their experience with prescribing or coordinating care for specialty medications that are distributed through an LDN; thus, sampling bias may have occurred.” I don’t think this is relevant for a qualitative study. In qualitative studies, bias cannot be avoided, they rather should be acknowledged. The concept of bias in qualitative and quantitative research is also different.

Don’t mix methods and results. Some of the methods section contain results, for example, lines 133 to 141 (including table 1).

What is the purpose of reporting the ethnicity/race of participants as well as their age and sex in Table 1? None of these characteristics was used to inform the data analysis. Perhaps, looking at the ethnicity/race of the participants, it appears that only practitioners from one ethnic group are recruited. Again, this is another example of missed opportunity by the researchers to include diverse perspectives!

Discussion

Please revise the limitation section of the discussion according to the above comments and suggestions.

Thank you.

7. PLOS authors have the option to publish the peer review history of their article (what does this mean?). If published, this will include your full peer review and any attached files.

Reviewer #1: **Yes: **Prof Greanious Alfred Mavondo

Reviewer #2: **Yes: **Dr Kebede Beyene

---

## [Author Response · Author response to Decision Letter 1]

14 Jul 2022

Response to Reviewers

Reviewer Comment Authors Response

Reviewer #1 

We have made comments to the Authors in our initial review where we had accepted the MS for publication. We appreciate the reviewer’s comments and previous feedback on the manuscript. 

Reviewer #2

Thank you for the opportunity to review this manuscript. Overall, the manuscript reads well and is easy to follow. Thank you for these kind words and feedback.

It is unclear how the authors ensured data saturation, which is an important process for deciding how many people to interview in qualitative studies We appreciate you identifying this omission. We’ve added to page 6, line 130 that data saturation was not assessed. Because there was a relatively small sample of providers/clinics able to be interviewed, we chose a priori to perform interviews/focus groups on all potential participants regardless of data saturation. 

It appears that the interviewee selections are mainly based on convenience rather than purposive sampling. Ten out of 14 participants (71.5%) are nurses/nurse practitioners. As such, the findings mostly represent nurses’ perspectives rather than the broader healthcare providers perspectives. The sampling was purposive as we used criteria that the clinic had to be familiar with and utilize LDDs in practice. The sample has a larger amount of nurses because they are typically the staff that address access issues for LDDs and therefore more knowledgeable on the topic. We’ve added to the limitations, page 28, line 586 that most of the participants were nurses, which contributes to limited generalizability.

Although clinical pharmacists are the main players/stakeholders in LDN/VSP, the authors failed to incorporate pharmacists’ perspectives in the study, which is a missed opportunity to better understand the complexity of the IHSSP and LDN systems. Also, it would have been better to incorporate manufacturers’ perspectives. Unless we hear all stakeholders’ views, it would be difficult to come up with workable solutions. We appreciate this perspective. The focus of this paper was to elucidate provider and clinic staff experience with LDNs rather than giving a sweeping view of the current LDN landscape. There is much more research to be done in this area as LDNs continue to increase and gathering opinions from pharmacists and manufacturers would be a great next step. We’ve added this to future directions (page 28, line 581).

It is unclear why the authors didn’t incorporate the field notes in their data analysis. We did not keep field notes for this study. Interviews were recorded and transcribed, providing the primary qualitative data. This is noted on page 9, line 155. We believe that the data provided meets PLOS One requirements of a minimal data set. 

“Member check” is an important step in ensuring the trustworthiness/quality of qualitative data. However, the authors didn’t provide an opportunity for the interview to comment on the findings or edit/confirm the interview transcripts. Thank you for this feedback. We have added this as a limitation to our methods (page 29, line 596). The experience of the Vanderbilt Qualitative Research Core is that providers are busy clinicians with minimal time and usually lack of desire to further review transcripts. However, we did not note this in the manuscript as this anecdotal experience may vary.

The focus group discussions only included two or three people. It is usually not recommended to have FGDs with less than five or six participants. This is a great point. We were unable to have larger focus groups due to limited staff with the knowledge/experience needed to speak to the topic. Additionally, we wanted to keep the focus groups specific to each clinic so that we could understand if certain clinical areas are affected differently than others, further limiting our potential to combine areas and enlarge our focus group size. We’ve added this as a limitation (page 29, line 591) 

The authors should also discuss the pros and cons of a non-health professional interviewer. Although having a non-health professional as an interviewer may help to avoid bias, there are several limitations of this approach in health research, and this should be properly discussed in the discussion section. We agree with the reviewer that this should be addressed and have added a discussion on pros/cons of this approach to the discussion (page 29, line 599)

Line 576-578: “Participants were individually selected by investigators based on their experience with prescribing or coordinating care for specialty medications that are distributed through an LDN; thus, sampling bias may have occurred.” I don’t think this is relevant for a qualitative study. In qualitative studies, bias cannot be avoided, they rather should be acknowledged. The concept of bias in qualitative and quantitative research is also different. Thank you for this feedback. We have updated the limitations to soften the language that that this recruitment method may have introduced limitation and changed verbiage to selection bias (page 29, line 595). 

Don’t mix methods and results. Some of the methods section contain results, for example, lines 133 to 141 (including table 1). We have moved the participant information to results as requested.

What is the purpose of reporting the ethnicity/race of participants as well as their age and sex in Table 1? None of these characteristics was used to inform the data analysis. Perhaps, looking at the ethnicity/race of the participants, it appears that only practitioners from one ethnic group are recruited. Again, this is another example of missed opportunity by the researchers to include diverse perspectives! We have retained providing the race/ethnicity, age, and sex to give reader’s insight into the makeup of participants as we believe this is typical in most qualitative research studies. As previously noted, we were limited in our ability to include diverse perspectives based on the small number of healthcare practitioners with robust experience with LDNs at our institution. We certainly would have liked to have a more diverse sample to engage. We’ve added the limited diversity in our sample to limitations (page 28, line 586)

Please revise the limitation section of the discussion according to the above comments and suggestions. We have revised limitations as noted above.

---

## [Editor Report · Decision Letter 2]

2 Aug 2022

Exploring healthcare providers' experiences with specialty medication and Limited Distribution Networks

PONE-D-21-28656R2

Dear Dr. Zuckerman,

We’re pleased to inform you that your manuscript has been judged scientifically suitable for publication and will be formally accepted for publication once it meets all outstanding technical requirements.

Kind regards,

Jonas Bianchi, DDD, MS, Ph.D

Academic Editor

PLOS ONE

---

## [Editor Report · Acceptance letter]

4 Aug 2022

PONE-D-21-28656R2 

Exploring healthcare providers' experiences with specialty medication and Limited Distribution Networks 

Dear Dr. Zuckerman:

I'm pleased to inform you that your manuscript has been deemed suitable for publication in PLOS ONE. Congratulations! Your manuscript is now with our production department. 

Kind regards, 

on behalf of

Dr. Jonas Bianchi 

Academic Editor

PLOS ONE